# LLMs for Sequential Optimization Tasks: from Evaluation to Dialectical Improvement

## Abstract

Large Language Models (LLMs) have demonstrated remarkable capabilities across diverse domains, opening new possibilities for solving complex optimization problems. This paper investigates the potential of LLMs as end-to-end designers for tackling Sequential Optimization Problems (SOPs), a challenging and pervasive class of tasks. To rigorously evaluate LLM performance, we introduce WorldGen, a dynamic benchmark for generating unseen SOPs with controllable complexity. Our initial findings show that while LLMs perform well on simpler SOPs, their effectiveness declines sharply as complexity increases. To address this, we draw inspiration from philosophical theories of reasoning—specifically, Hegelian Dialectics—and propose ACE, a dialectical framework that enhances LLM performance in SOPs without requiring retraining or fine-tuning.

## 1 Introduction

**Reframing Optimization in the Age of AI:** Optimization lies at the heart of decision-making across disciplines—from engineering and economics to healthcare and scientific discovery. The essence of decision-making lies in choosing the best option from a set of alternatives, driven by objectives such as efficient resource allocation, cost minimization, profit maximization, or performance enhancement of systems and infrastructures Chong & Zak (2013). Traditionally, solving optimization problems has required domain expertise, algorithmic craftsmanship, and careful handling of challenges such as high dimensionality, nonlinearity, and stochasticity Datta et al. (2019). Despite decades of progress, the process remains labor-intensive and brittle, often demanding manual tuning and heuristic design. Consequently, there is a continuous quest across various fields to simplify and automate the optimization process.

**A New Opportunity:** Recent advances in LLMs, such as GPT-4 OpenAI (2023), have sparked interest in their potential to automate aspects of optimization Song et al. (2024). However, most existing approaches treat LLMs as auxiliary components—constructing surrogate models, guiding heuristics, or assisting within human-designed pipelines Wang et al. (2024); Liu et al. (2024); Lange et al. (2024). In contrast, in this paper, we ask a fundamental question: How proficient are LLMs at performing *end-to-end design* for solving Sequential Optimization Problems (SOPs)?

**Assessing LLMs in a New Context:** Although various benchmarks exist to evaluate LLM performance in general and specialized tasks (e.g., coding, mathematics), their capabilities in solving optimization problems, particularly SOPs, remain underexplored. SOPs involve making a series of decisions over time, where each decision affects subsequent options and outcomes, creating a complex web of interdependencies. Different challenges contribute to this issue, primarily the need for defining a representative set of SOPs and ensuring that observed performance is not influenced by data contamination or prior exposure during the LLMs' training Dong et al. (2024); Mialon et al. (2023); Chen (2023).

**On-Demand SOP Generation:** Motivated by these opportunities and challenges, first, we aim to address this research question: *How can we evaluate the performance of LLMs in SOPs?* We introduce a straightforward yet effective framework, WorldGen, capable of generating unseen SOPs with controllable complexities on demand. WorldGen contrasts with most existing static benchmarks (e.g., MMLU Hendrycks et al. (2021), GLUE Wang et al. (2018), SuperGLUE Wang et al. (2019), GSM8k Cobbe et al. (2021), etc.) which become obsolete as LLMs evolve Mialon et al. (2023). Utilizing this dynamic framework, we have made two key observations: (1) For relatively simple

optimization tasks, with a single global maximum and no local maxima (i.e., simple surfaces and scenarios), current LLMs can design solutions that solve them efficiently. (2) As the complexity of the optimization problems increases, the performance of LLMs degrades significantly and becomes unsatisfactory.

**Improving LLMs with Roots in Philosophy:** Next, inspired by these observations and the poor performance of LLMs in SOPs, we propose a systematic approach to enhance the performance of off-the-shelf LLMs (in the context of SOPs) without necessitating any retraining, treating the LLM as a black box. In particular, we propose ACE, to demonstrate that LLMs can be transformed from passive assistants into active designers of optimization strategies. ACE empowers LLMs to select, orchestrate, and adapt optimization methods dynamically—treating SOPs as open-ended design challenges rather than scripted tasks. To support this, ACE introduces a principled dialectical reasoning process inspired by Hegelian Dialectics Hegel (1807; 1812). Instead of relying on prompt engineering, ACE structures inference through *thesis–antithesis–synthesis* cycles, enabling iterative refinement and strategic exploration. Crucially, ACE allows LLMs to go beyond generating the next query point—it enables them to explore alternative strategies, adapt to feedback, and evolve their approach across the entire optimization lifecycle (See Appendix A for a brief background in philosophy on reasoning, dialectics, and Hegel's framework).

**Main Contributions:** In summary, this paper makes the following key contributions:

1. We design WorldGen, a framework to assess the performance of LLMs in SOP settings, addressing the data contamination issues typically associated with general LLM benchmarks (detailed in section 3.1). WorldGen allows for the growth of evaluation complexity in line with the advancement of LLMs.
2. Using this framework, we provide initial observations on the poor performance of current LLMs in SOPs, motivating the need for structured reasoning frameworks that support full-cycle design (detailed in section 3.4).
3. We present ACE, a novel approach inspired by one of the most successful dialectical hypotheses in philosophy for explaining and enhancing reasoning. We show that ACE improves the performance of LLMs in end-to-end optimization design through dialectical reasoning. This enhancement occurs at test time, without any retraining or fine-tuning (detailed in section 3.5).

## 2 RELATED WORK

Our work intersects with several research areas: (1) LLM-Assisted Optimization, (2) Prompt Engineering, (3) Benchmarks for Evaluating LLMs, and (4) Multi-Agent Systems. Here, we briefly review the first two areas and provide a more detailed discussion of the latter two in Appendix B.

**LLMs for Optimization:** Recent research on leveraging LLMs for optimization can be broadly grouped into two categories:

1. *LLMs as components in human-designed optimization procedures:* In this paradigm, LLMs serve as *auxiliary* tools within pre-defined algorithms, assisting with tasks such as generating initial candidate solutions, constructing surrogate models, or guiding search heuristics. Examples include integrating LLMs into evolutionary strategies or Bayesian optimization (BO) pipelines, where human intervention remains essential for prompt design, heuristic tuning, and algorithmic orchestration Wang et al. (2024); Liu et al. (2024); Lange et al. (2024). While these approaches demonstrate that LLMs can complement traditional optimizers, they limit the model to a supporting role rather than granting full desinger autonomy.
2. *LLMs as black-box solvers:* A smaller body of work explores using LLMs to directly generate queries Yang et al. (2024). However, the design choices in this group remain largely confined to the few-shot learning capabilities of LLMs (e.g., generating the next query point based on list of previous ones as in Yang et al. (2024)). Even in this setting, performance often depends on manual heuristics—such as specifically sorting previous samples or crafting specialized prompt templates—to stabilize behavior and improve convergence. This reliance on human-designed scaffolding constrains the autonomy and generality of these approaches and reduces their true "black-box" nature.

Our framework, ACE, departs from both paradigms in three key ways. First, ACE positions the LLM as an *autonomous designer*, capable of independently selecting and applying optimization strategies

(e.g., BO, Genetic Algorithms, etc.) without human intervention or guidance. Unlike prior work that embeds LLMs into rigid pipelines, ACE enables end-to-end decision-making, allowing the model to hypothesize, test, and refine strategies dynamically. Second, ACE eliminates heuristic prompt engineering by introducing a *principled dialectical reasoning process* at inference time. Rather than relying on ad hoc tweaks, ACE structures reasoning through thesis–antithesis–synthesis cycles, inspired by Hegelian Dialectics, to improve solution quality systematically. Finally, ACE emphasizes *true autonomy*: the model is not restricted to generating the next query point from a curated history but can explore alternative strategies and adapt its approach based on feedback.

**Evaluation Beyond Static Benchmarks:** Existing evaluations of LLM-based optimization often rely on public datasets or well-known problem suites (e.g., Traveling Salesman Problem Applegate et al. (2006), Black-Box Optimization Benchmarking Finck et al. (2009)), which risk contamination from pretraining corpora and may inadvertently measure memorization rather than reasoning. To address this, we introduce WorldGen, a dynamic framework for generating unseen SOPs with controllable complexity. Unlike static benchmarks, WorldGen can evolve alongside model capabilities, ensuring fair and forward-looking assessment of LLM performance in genuinely novel scenarios.

**Prompt Engineering (PE):** Since we treat LLMs as black-boxes and focus on improving the off-the-shelf models without any retraining, it will be natural to mention works in PE domain here. Prompting techniques have become essential in enhancing the performance and versatility of LLMs. These techniques range from Zero-shot Prompting, where models are given tasks without prior examples, to Few-shot Prompting, which provides a few examples to guide responses Brown et al. (2020). Chain-of-Thought encourages models to generate intermediate reasoning steps Wei et al. (2022), Self-Consistency (Majority Vote) generates multiple outputs to select the most consistent one Wang et al. (2022); Lewkowycz et al. (2022), and Generate Knowledge Prompting prompts the model to produce relevant background information before answering Liu et al. (2021). Tree of Thoughts structures reasoning as a tree to explore different branches Yao et al. (2024); Long (2023). Retrieval Augmented Generation combines document retrieval with generation for improved accuracy Lewis et al. (2020). Automatic Prompt Engineer uses algorithms to refine prompts Zhou et al. (2022), while Active-Prompt Diao et al. (2023) adjusts prompts based on performance feedback and Program-Aided Language incorporate programming logic Gao et al. (2023). Techniques like ReAct combine reasoning and acting steps Yao et al. (2022), and Self-Reflection prompts models to reflect on their responses for better outcomes Madaan et al. (2024); Shinn et al. (2024). These diverse techniques collectively enhance the adaptability and effectiveness of LLMs. ACE is orthogonal compared to these techniques and as we later show, it can be combined with them (detailed in section 3.5).

## 3 ASSESSING & IMPROVING LLMS IN SOPS

SOPs are pervasive across diverse domains, ranging from logistics and resource allocation to machine learning and operations research. Requiring specialized knowledge to address practical issues such as high dimensionality, nonlinearity, and the dynamic, unpredictable nature of real-world settings make SOPs complex in their nature Datta et al. (2019). Automating the solution of such problems is highly desirable, as it can lead to efficiency gains and innovative solutions to complex challenges. On the other hand, LLMs have demonstrated remarkable capabilities, including proficiency in coding and exceptional context-awareness, making them promising candidates for tackling SOPs.

So, naturally, exploring the performance of LLMs in addressing these crucial tasks is a significant step forward in understanding their broader applicability and the opportunity to solve these problems automatically. However, evaluating LLMs in this context comes with it own set of challenges. Key concerns include managing data contamination, ensuring that the problems and their solutions were not inadvertently exposed during training, and consequently distinguishing between genuine reasoning and mere memorization. Additionally, access to a set of representative optimization problems is vital to effectively assess LLMs' capabilities in solving sequential tasks. Addressing these challenges will enable a deeper understanding of the role LLMs can play in advancing optimization methodologies.

### 3.1 WORLDGEN

**Core Idea:** At the heart of optimization lies the task of finding the optimum point(s) in an $n$-dimensional world, mathematically expressed as $f(x_1, x_2, \ldots, x_{n-1})$. So, instead of focusing on

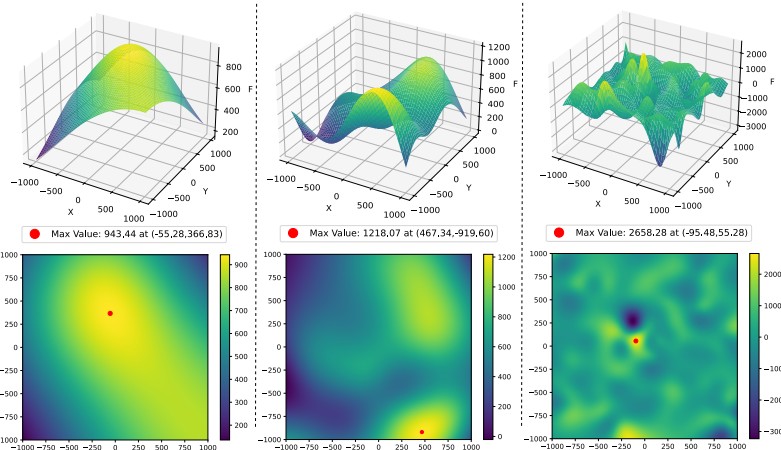

Figure 1: Samples of generated 3D worlds (and their 2D heatmap versions) for different SOP complexities: very simple (left), simple (middle), and medium (right).

specific optimization problems, we shift our attention to the worlds that represent these problems. In other words, rather than trying to come up with specific optimization problems, which may inadvertently introduce biases or contamination from training data, we focus on directly generating $n$-dimensional worlds that can represent the solution spaces for a wide range of SOPs. That said, we do not predefine the optimization problem. Instead, the task naturally emerges as finding the maximum (or other extrema) in the generated $n$-dimensional world. This setup allows us to define flexible problems while preserving the integrity of the test environment.

**Benefits and Advantages:** This approach ensures that neither the optimization problem nor its solution was exposed to the LLM during training. By doing so, we can mimic a real-world scenario where an optimization expert is asked to tackle a newly faced optimization problem using any techniques or strategies they prefer. Utilizing this approach brings some advantages. It offers generative flexibility, allowing the $n$-dimensional world to represent an infinite variety of optimization problems, from simple to highly complex ones. By abstracting the problem into a generated world, it ensures unbiased evaluation, reducing contamination from known problem-solution pairs and providing a more acceptable measure of the LLM's capability. Our world generator, WorldGen, enables the creation of increasingly complex worlds that test the limits of learning agents and provides a platform for benchmarking them under controlled yet dynamic conditions. Figure 1 shows samples of generated 3-D worlds with different complexity levels.

### 3.2 LLM AND ACCESSING THE WORLD

**An Interactive Cycle:** To enable the LLM agent to perform its task effectively, we provide it with access to the generated world through an interactive cycle as shown in Figure 2. This ensures a dynamic sequential process where the agent iteratively learns and refines its approach based on the information it gathers. Simply put, in each iteration, the LLM agent is allowed to interact with the world by selecting a batch of interested points where each point is a vector, $v_i$ of size $n - 1$. Then, the world responds by revealing the corresponding values of $f(v_i)$ to the LLM agent. This will end one iteration/round of the interaction. In the next round, the LLM utilizes this feedback to determine the next set of points to query.

**Supporting Coding & Providing Flexibility:** Due to the complexity of SOPs, the LLM agent is permitted to provide a Python code as part of its response and therefore utilize any library it deems necessary for solving the problem. This freedom ensures that the agent can employ a diverse range of tools and techniques to explore and analyze the generated world. As part of its role, the World is responsible for executing the Python code generated by the LLM agent. Once the code is executed, the World provides the results back to the agent as part of its feedback, enabling it to adapt and refine its strategy in subsequent iterations. In case of errors that may arise during execution, the World is responsible to provide the details of the errors and return meaningful feedback. Moreover, the LLM agent is not constrained by a fixed set of queries or techniques. Instead, it has the freedom to decide

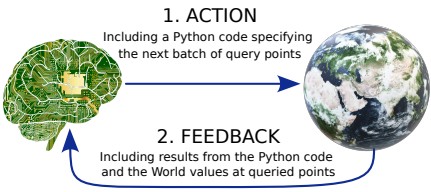

Figure 2: Interactive cycle for a LLM agent & the World

Table 1: The success rates of LLM$^\star$ (GPT-5-chat as base model) in worlds with diff. complexity levels

| WORLD COMPLEXITY | SUCCESS RATE ↑ |
|---|---|
| L0 (VERY SIMPLE) | 100% |
| L1 (SIMPLE) | 60% |
| L2 (MEDIUM) | 44.5% |

how to approach the problem, including leveraging mathematical models, heuristics, or machine learning techniques. This flexibility mimics the conditions under which human experts operate when solving optimization problems. The interaction between the LLM and the World creates a real-time feedback loop. The LLM continually refines its understanding of the world based on the revealed data, while the World executes the agent's strategies and provides results.

## 3.3 NOTION OF EFFICIENCY

**To Solve or to Efficiently Solve:** Exhaustively searching through all possible solutions, or brute force, is always a solution to any optimization problem; however, it represents the most inefficient approach. Thus, merely solving an optimization problem is not the primary goal; solving it efficiently is what truly matters. To formalize this, we require the notion of an efficient solution. But how can we define efficiency in a meaningful and practical way here?

**The Expert Solution:** To address this question, we designed a baseline referred to as the Expert Solution. This baseline serves as a reference point for assessing the efficiency of the LLM agent's performance. The Expert Solution is crafted using different optimization techniques, including a combination of Monte Carlo search methods, Bayesian optimization, and Active Learning strategies. An important aspect of the Expert Solution is the introduction of a *query budget*. This budget represents the number of queries required by the Expert Solution to reliably solve the optimization problem. It provides an upper bound on the number of interactions with the environment that are necessary to achieve a solution. That said, alongside the optimization problem, the LLM agent is provided with the query budget and instructed to not only solve the optimization problem but also do so within the given query budget.

**Incentivizing Efficiency:** This setup ensures that the agent is incentivized to prioritize efficiency. It must strategize its queries, balancing exploration and exploitation to maximize the information gained from each interaction. By enforcing a query budget, we can objectively evaluate LLM's efficiency and effectiveness in solving the problem. Ultimately, the notion of efficient solutions pushes the LLM agent beyond simple problem-solving, encouraging it to adopt creative and resource-conscious strategies that align with real-world optimization challenges.

## 3.4 EVALUATING LLMs PERFORMANCE IN SOPs

**The Setup:** To evaluate the performance of the LLM agent, we follow a structured approach based on repeated experiments. We begin by generating worlds, characterized by a complexity index. In particular, to simplify experiments, visualizations, and keep overall token usage manageable, we focus on 3-D worlds and three levels of complexity: very simple (L0), simple (L1), and medium (L2). Next, we apply the Expert Solution to solve the corresponding SOPs associated with the generated world. As a result, we find a query budget required to achieve this reliably. Then, LLM agent is asked to solve the problem constrained by the query budget. We repeat each trial 10 times and define the success rate of the LLM agent as the proportion of runs in which it successfully identifies the optimal solution within the given query budget.[1] For each complexity category, we compute the success rate for every individual World, then average these values across all Worlds in that category to obtain the final success rate. This metric provides a quantitative measure of the agent's effectiveness and efficiency. In these experiments, we use the GPT-5-chat model as a strong recent baseline.[2]

**The Default Scheme:** Without involving any prompting techniques, the success rates become very low (close to 0%). Therefore, we borrowed ideas from few-shot learning Brown et al. (2020), Chain of

---

[1]We apply a relaxation criterion by treating any value within 5% of the optimum as the optimal point.
[2]ACE code is available at Anonymous (2025).

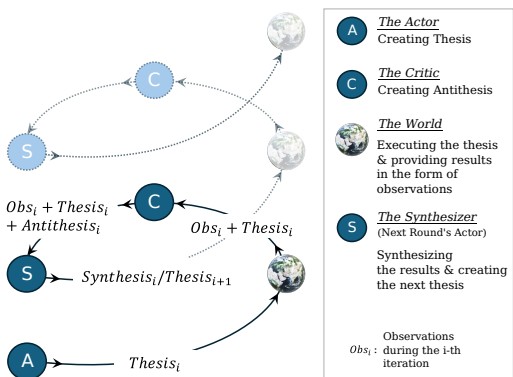

Figure 3: ACE and the spiral of thoughts

Thought (CoT) Wei et al. (2022), and added other techniques such as proper role assignment Karpathy (2023) to improve the performance of the LLM agent. We name the resulting scheme *LLM\** and treat it as our default scheme from now on. In particular, the prompt given to the LLM agent includes: [Role Assignment], followed by [Problem Definition & Examples], [General Helpful Notes], and [Required Response Format]. The [Required Response Format] itself consists of [Plain Description of Current Strategy], [Python Code Implementation of the Strategy], and [Maximum Value Found So Far] fields (check Appendix E for more details).

**Results:** Table 1 summarizes the results of LLM\* in various scenarios. These results optimistically suggest that LLMs understand SOP settings and are familiar with optimization techniques. As expected, the success rate of LLM\* depends on the complexity of the underlying world. However, even in relatively simple scenarios (L1), LLM\* does not achieve a high success rate. In straightforward settings where the world representing the SOP has only one global optimum with no other local optimum points, LLM\* effectively utilizes general optimization techniques such as gradient ascent. But, when the world exhibits some complexity, with a few local optimum points in non-trivial parts of the space—common in real-world scenarios—LLMs struggle to strategize properly and find the global optimum (check Appendix C for more details).

### 3.5 A DIALECTICAL PERSPECTIVE TO ENHANCE LLMS IN SOPS

Motivated by the unsatisfactory performance of vanilla LLMs in non-straightforward SOPs, we aim to address a natural follow-up question: Can we enhance the performance of LLMs without relying on retraining, fine-tuning, or post-training modifications?

**The Core Idea:** To that end, we propose a framework grounded in the principles of Hegelian Dialectics, offering a formal structure for enhancing LLM performance through dynamic, iterative reasoning. Both sequential optimization and dialectical reasoning share a foundational trait: *they operate through structured cycles that aim to progressively improve outcomes*. This shared emphasis on iterative refinement provides a natural bridge between optimization strategies and dialectical logic. Using the terminology of Hegelian Dialectics, we conceptualize a general LLM agent as a *Thesis Generator*—a module that observes a problem and produces an initial solution or response.[3] However, we argue that relying solely on a Thesis Generator is insufficient for solving SOPs. A more robust architecture should incorporate two additional components: an *Antithesis Generator*, which analyzes gaps and challenges in the initial output, and a *Synthesis Block*, which integrates insights to produce refined solutions. Together, these three components form a dialectical reasoning cycle, as formally described by Hegel Hegel (1807; 1812), enabling the system to iteratively evolve and improve its responses.

**ACE** [4]**:** The Antithesis Generator plays a critical role by challenging the solutions produced by the Thesis Generator. It identifies potential flaws, contradictions, or alternative perspectives that may have been overlooked. This counterbalance forces the system to evaluate its assumptions critically and consider a broader range of possibilities. The Synthesis Block then reconciles the Thesis and

---

[3]This abstraction remains valid even within Minsky's multi-agent paradigm Minsky (1988), where a complex problem is decomposed into subproblems handled by specialized agents that communicate and collaborate.

[4]ACE stands for **A**ct, **C**ritique, and **E**volve

Table 2: Success rates of schemes across base LLMs and complexity levels in 3D worlds

| Base Model | Level | Scheme | | | | |
|---|---|---|---|---|---|---|
| | | LLM* | Self-Reflection | Debate | Majority | ACE |
| Kimi-K2 | L1 | 39.2 | 44.5 | 36.4 | 47.5 | **53.3** |
| | L2 | 18.2 | 24.5 | 18.2 | 16.4 | **30.0** |
| DeepSeekV3 | L1 | 65.0 | 53.3 | 60.0 | 45.0 | **75.0** |
| | L2 | 60.9 | 52.7 | 53.6 | 28.2 | **68.2** |
| GPT-5-chat | L1 | 60.0 | 70.0 | 70.0 | 64.2 | **70.8** |
| | L2 | 44.5 | 51.8 | 50.0 | 47.3 | **59.1** |
| GPT-4.1 | L1 | 56.7 | 60.8 | 50.0 | 55.8 | **64.2** |
| | L2 | 48.2 | 36.4 | 34.5 | 32.7 | **50.0** |
| GPT-4o | L1 | 46.7 | 42.2 | 50.8 | 48.3 | **53.8** |
| | L2 | 43.6 | 39.1 | 44.5 | 24.5 | **49.1** |
| GPT-4o-mini | L1 | 40.0 | 37.5 | 35.8 | 41.7 | **51.7** |
| | L2 | 21.8 | 25.5 | 24.5 | **32.7** | 24.5 |
| O3-mini | L1 | 10.0 | 10.0 | 15.0 | 4.2 | **31.7** |
| | L2 | 8.2 | 3.6 | 7.3 | 1.8 | **24.5** |

Antithesis, combining their insights to produce a more refined and coherent solution. This iterative interplay between Thesis, Antithesis, and Synthesis ensures that the system continuously evolves its understanding and response, ultimately arriving at a more suitable outcome. This dialectical structure led us to introduce our solution, *ACE*, embodying three components: (1) Actor, (2) Critic, and (3) Synthesizer. The relationship between these components is shown in Fig. 3. The iterative cycle of solving a problem starts with the Actor creating an initial thesis. This thesis is then implemented and executed in the world, and the corresponding outcomes and results (called observations) are gathered. Next, the Critic examines the initial thesis and the corresponding observations to generate an antithesis. The thesis, antithesis, and corresponding observations are then fed into the Synthesizer. The Synthesizer creates an evolved thesis, completing an iteration/round. The cycle continues by treating the evolved thesis as the next initial thesis in the cycle. As we later show in section 4, ACE significantly improves the performance of LLMs with no modification to their architecture and no extra post-training or fine-tuning. By embedding a dialectical reasoning process into the system, ACE creates a more adaptive process that is better equipped to handle complex and nuanced sequential optimization tasks (Appendix D provides samples and details of ACE's dialectical process).

## 4 EVALUATION

### 4.1 OVERALL RESULTS

**Settings:** To put the performance improvements of ACE in proper context, we implemented several recent related proposals including Self-Reflection Madaan et al. (2024), Majority Vote Wang et al. (2022); Lewkowycz et al. (2022), and Debate Du et al. (2024) and compared them with ACE. To have a fair comparison, for the Majority and Debate schemes we set the total number of agents to three and two agents, respectively, to roughly match the token usage of ACE. Later, in section 4.2, we perform more evaluations with higher number of agents for them. We accompany all these schemes with the additional prompting techniques appeared in LLM* to have a fair comparison (check Appendix E for more details). As for the LLMs, we use 7 different models: DeepSeekV3 DeepSeek-AI (2024), GPT-5-chat OpenAI (2025a), GPT-4.1 OpenAI (2024b), GPT-4o OpenAI (2024a), GPT-4o-mini OpenAI (2024c), O3-mini OpenAI (2025b), and Moonshot's Kimi-K2 AI (2025). We repeat evaluations 10 times and report the overall success rate of different schemes in each category.

**Results:** Table 2 summarizes the findings and confirms two things. First, WorldGen successfully scales difficulty: all models show a clear drop from L1 to L2, highlighting the challenge of sequential optimization as complexity grows. Second, ACE consistently improves performance across models and levels, remaining the best or tied for best in nearly every scenario. This holds for strong models like DeepSeekV3 and GPT-5-chat, where ACE pushes success rates to the top tier (e.g., DeepSeekV3 L1: 75.0%), and for smaller models like O3-mini, where ACE lifts performance from small baselines to meaningful success (e.g., L2: 24.5%)[5]. ACE operates by treating LLM as a black-box, relying on

---

[5]A sample dialectical process in ACE is shown in Appendix D

Table 3: Averaged total consumed tokens of schemes (with O3-mini as base) in L1 class

| SCHEME | NORMALIZED ↓ | TOTAL TOKENS ↓ |
|---|---|---|
| LLM* | 1 | 5659 |
| SELF-REFLECTION | 1.88× | 10660 |
| ACE | 2.85× | 16146 |
| DEBATE | 3.48× | 19710 |
| MAJORITY | 3.75× | 21260 |

Table 4: Success rates and avg normalized total tokens (to LLM*'s tokens) in L2 3D worlds

| SCHEME | O3-MINI | | GPT-4.1 | |
|---|---|---|---|---|
| | SUCCESS ↑ | COST ↓ | SUCCESS ↑ | COST ↓ |
| LLM* | 8.2 | 1 | 48.2 | 1 |
| DEBATE* | 9.1 | 31.97× | 35.5 | 30.89× |
| MAJORITY* | 0.0 | 6.06× | 37.3 | 6.90× |
| ACE | **24.5** | **3.78×** | **50.0** | **2.87×** |

the existing abilities of the LLM without retraining or modifying its weights. So, its performance naturally varies according to the base model's capabilities in SOP setting. While the absolute performance varies, the pattern is stable: ACE works with different models with different starting capability. Among the models tested, the only exception is GPT-4o-mini at L2, where Majority slightly leads, suggesting that when base ability is very low and query budgets are tight, majority can occasionally outperform a single dialectical chain.

**Takeaway:** ACE is the most reliable scheme across 13 of 14 settings, improving strong models and transforming weaker ones on harder tasks. Its stepwise dialectical loop prevents error cascades and adapts to complexity without retraining, making it a robust choice for sequential optimization under fixed budgets.

## 4.2 DEEP DIVE

**Cost Comparison:** We evaluate the cost of using ACE by analyzing the total number of tokens consumed and comparing it with other approaches. Table 3 reports the average total token usage across experiments in L1 scenarios for various schemes, along with their corresponding values normalized to the default single-agent baseline, LLM*, using O3-mini as the base model. On average, ACE consumes $2.87\times$ total tokens compared to LLM*; however, it remains more efficient than multi-agent schemes such as Debate and Majority, which incur significantly higher token costs.

**ACE vs. Multi-Agent Schemes with More Agents:** A natural assumption might be that increasing the number of agents in schemes like Debate and Majority would enhance performance. So, we conducted additional experiments, scaling the number of agents to seven (referred to as Debate* and Majority*) and comparing their performance and token costs with ACE. These experiments were carried out using GPT-4.1 and O3-mini as base models in the L2 scenarios. The results are summarized in Table 4. As shown in Table 4, increasing the number of agents in these schemes dramatically increases cost—up to $32\times$ the baseline—while offering no meaningful performance benefit; in some cases, success rates even decline. Schemes like Debate*, which require exchanging responses among all agents in each round, can experience exponential growth in token consumption as the number of agents increases. This inefficiency is particularly evident in SOP settings, where sequential decision-making processes require multiple rounds of interaction, as illustrated by the token cost data in Table 4. The key takeaway is that simply increasing the number of agents in tasks involving sequential decision-making, such as those in SOP settings, does not necessarily yield better results. Instead, it often introduces inefficiencies and performance degradation in solutions like Debate* and Majority*.

## 5 LIMITATIONS & A BRIEF DISCUSSION

**WorldGen's Limitation:** WorldGen effectively generates worlds with adjustable complexity for testing LLMs in SOPs, but relies on manually designed Expert solutions to solve the SOP. This dependence on human expertise for robust baselines can be time-intensive and limit the automation potential of the approach. We leave addressing the fully automated objective to future work.

**Limitations of ACE:** ACE's dialectical framework has demonstrated great performance in our main targeted domain, SOPs, but its effectiveness in other domains, particularly those lacking real-time feedback, remains an open question. In static question-answering or static tasks without iterative refinement, the benefits of ACE may be limited. Additionally, by treating LLMs as black boxes, ACE's performance is inherently bound by the capabilities of the underlying model. Moreover, while its token consumption is lower than multi-agent schemes like Debate or Majority, ACE incurs a slight overhead compared to single-agent approaches. This trade-off is minor in complex SOP tasks but could pose challenges in resource-constrained scenarios.

**Is ACE better than BO, reinforcement learning (RL), etc?** This question reflects a common misconception about our goal. We do not aim to propose a new optimization algorithm or to test whether LLMs can invent one. Instead, we position LLMs as *designers*—similar to human optimization experts—who can draw on any algorithm in their toolbox. From this angle, comparing ACE to BO, RL, or other techniques is irrelevant, since LLMs can already invoke, adapt, or combine such methods during problem solving (e.g., Appendix D demonstrates examples of models utilizing well-known techniques such as BO to solve SOPs). The key point is not whether ACE is superior to classical algorithms, but that when LLMs are placed in the designer's seat for end-to-end SOP solving, they benefit from structured scaffolds like ACE's dialectical process, which amplifies their ability to reason, critique, and refine solutions within the interactive, stepwise SOPs.

**On the Potential of ACE:** The potential of ACE, rooted in its Hegelian dialectical framework, extend beyond solving SOPs. Its dialectical approach, mirroring human-like problem-solving processes, fosters solutions that are not only accurate but also deeply contextual and well-reasoned. Furthermore, Hegelian philosophy provides a foundation to explain the effectiveness of other prompt engineering techniques, such as self-reflection, by framing them within a structured dialectical process. This perspective can deepen our understanding of existing methods and their mechanisms. Additionally, the Hegelian-inspired framework offers a powerful structure for *generating synthetic data*. Its iterative nature facilitates the creation of diverse, high-quality datasets that reflect a broad range of perspectives and solutions, making them invaluable for training and fine-tuning LLMs to tackle complex and nuanced tasks effectively.

**LLM$^\star$ Could Have Been Better!** A fair criticism might be that LLMs might perform better in solving SOPs with improved prompt engineering. We are not claiming that LLM$^\star$ represents the optimal default scheme; rather, we argue that it serves as a robust baseline. Even with carefully designed prompts, LLMs' performance in this setting remains limited, highlighting the need for approaches like ACE to unlock their full potential and deliver superior performance.

**What If the Next LLM Becomes Very Capable?** A more capable LLM makes ACE even more useful, not less. As demonstrated in Section 4, a better base model serves as a stronger foundation, enabling more performance improvements. In essence, ACE with its dialectical base is designed to complement and amplify the capabilities of any LLM, regardless of its initial proficiency in SOP context. By leveraging ACE, we can transform an already impressive LLM into an extraordinary one, pushing what is possible and unlocking new levels of performance in this domain.

**Dialectics vs. Debate:** From the philosophical point of view, debate is competitive, aiming to persuade an audience of one position's superiority. While effective in contexts like politics or law, it often sacrifices deeper inquiry for rhetoric and winning. Dialectics, however, fosters a cooperative approach, treating opposing perspectives as opportunities for growth. Through structured dialogue, dialectics seeks deeper truths, as seen in the Socratic and Hegelian methods, encouraging intellectual humility and a shared pursuit of wisdom. While debate has been significant in philosophical traditions, figures like Socrates criticized its focus on persuasion over truth. Dialectics, with its emphasis on dialogue and synthesis, is regarded as superior for fostering intellectual growth.

# 6  FINAL NOTE

Our exploration into the capabilities of LLMs in tackling sequential optimization problems has revealed both their potential and their current limitations. Through the development and use of WorldGen, we have shown that while LLMs exhibit impressive abilities, they still face challenges with even relatively simple SOPs. These findings have led us to propose a novel approach inspired by philosophical reasoning frameworks, aiming to enhance LLM performance in innovative yet easy to reason about ways. We believe that this work can open new avenues, encouraging the integration of philosophical reasoning frameworks into AI systems. By fostering a deeper understanding and application of these frameworks, we can pave the way for more robust and intelligent systems. We hope our efforts inspire others to explore these interdisciplinary approaches, ultimately contributing to the advancement of LLMs and its applications across diverse fields.[6]

---

[6]In accordance with ICLR guidelines, we acknowledge that LLMs were used for proofreading and polishing the text of this paper.

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

## A BACKGROUND: REASONING AND DIALECTICS

**Reasoning:** Despite considerable achievements, LLMs' reasoning capability continues to be a subject of intense debate within the AI research community. A key challenge lies in reaching a consensus on what reasoning entails, how it should be defined, and how it can be reliably measured. Interestingly, the concept of reasoning is not new. The domain of philosophy has a rich tradition of exploring and formalizing reasoning through centuries of discourse Aristotle (-350; -340); Plato (-380); Descartes (1641); Hume (1739); Kant (1781); Mill (1843); Hegel (1807); Nietzsche (1886); Wittgenstein (1921);

Heidegger (1927); Popper (1934); Kuhn (1962); Adorno (1966). From ancient philosophers such as Aristotle, who developed formal logic as a foundation for reasoning Aristotle (-350; -340), to more recent thinkers like Hegel, who introduced dialectics as a dynamic framework for understanding processes of thought Hegel (1812; 1807), the philosophical study of reasoning has produced a wide range of influential theories and formal systems. These works not only define reasoning but also provide structured frameworks for improving and analyzing it.

**Dialectics:** As a method of reasoning and philosophical argumentation, dialectics involves the resolution of contradictions through a process of development and transformation. Rooted in ancient philosophy, dialectics was first formalized by thinkers like Socrates and Aristotle, who used it as a tool for logical inquiry. Over time, dialectics evolved into a broader philosophical framework, describing the dynamic process through which contradictions are identified, explored, and resolved Hegel (1807); Engels (1875). At its core, dialectical thinking posits that reality is composed of opposing forces or contradictions, and that these contradictions are not static but dynamic, evolving over time. The resolution of these contradictions leads to the emergence of new, higher forms of understanding or being.

**Hegelian Dialectics:** Introduced by the German philosopher Georg Wilhelm Friedrich Hegel, Hegelian Dialectics crystallizes the modern notion of dialectics by proposing a structured process of development through three stages: *thesis*, *antithesis*, and *synthesis* Hegel (1812; 1807). The thesis represents an initial idea or condition, the antithesis introduces a contradictory or opposing force, and the synthesis resolves the tension by merging elements of both into a higher, more comprehensive understanding. Hegel viewed this triadic process as the driving force of intellectual, historical, and societal progress, emphasizing that contradictions, which he calls "negations", are not merely obstacles but necessary components of growth and transformation. His dialectical framework has had profound influence across disciplines, from philosophy to political theory.

**Dialectics vs. Debate:** From the philosophical point of view, debate is competitive, aiming to persuade an audience of one position's superiority. While effective in contexts like politics or law, it often sacrifices deeper inquiry for rhetoric and winning. Dialectics, however, fosters a cooperative approach, treating opposing perspectives as opportunities for growth. Through structured dialogue, dialectics seeks deeper truths, as seen in the Socratic and Hegelian methods, encouraging intellectual humility and a shared pursuit of wisdom. While debate has been significant in philosophical traditions, figures like Socrates criticized its focus on persuasion over truth. Dialectics, with its emphasis on dialogue and synthesis, is regarded as superior for fostering intellectual growth.

In this work, inspired by Hegel's well-established framework, we demonstrate how the capabilities of LLMs can be enhanced by adapting dialectics.

## B    RELATED WORK: PART II

**Benchmarks for Evaluating LLMs:** There are numerous benchmarks for evaluating LLMs, ranging from general-purpose (e.g., GLUE Wang et al. (2018), SuperGLUE Wang et al. (2019), ARC Clark et al. (2018), HellaSwag Zellers et al. (2019), BIG-bench Srivastava et al. (2022), GAIA Mialon et al. (2023)) to domain-specific (e.g., FinBen Xie et al. (2024) for finance, LegalBench Guha et al. (2024) for legal reasoning, GSM8K Cobbe et al. (2021) and MATH Hendrycks et al. (2021) for mathematical reasoning, HumanEval Chen et al. (2021) and MBPP Austin et al. (2021) for coding, MultiMedQA Singhal et al. (2023) for healthcare, etc.) and ones requiring professional level knowledge in various fields such as law or science (e.g., MMLU Hendrycks et al. (2021)). Our framework, WorldGen, falls into the domain-specific category. It addresses the issue of being static and become obsolete with the rapid advancements in LLMs by providing a dynamic tool for generating SOPs with varying controllable complexity.

**Multi-Agency:** Minsky was among the early pioneers to introduce the idea of multi-agent systems Minsky (1988). His notion of multi-agency involves dividing complex cognitive tasks into smaller parts, delegating them to specialized "agents", and integrating the results into a coherent solution. Inspired by Minsky's vision, recent works have utilized and implemented multi-agency in LLM-based systems. Some focus on building general infrastructures for autonomous cooperation among communicative agents (e.g., CAMEL Li et al. (2023) and AutoGen Wu et al. (2023)). Others focus on specific multi-agent solutions or tailored applications. For instance, MetaGPT Hong et al.

---

**Strategy in Round 1 (Global Exploration + Plan)**

Phase-1 coarse grid (Latin Hypercube) to discover "interesting" regions while avoiding early clustering. Phase-2 zoom: once any value $> 0.8 \cdot$ max_seen is observed, launch a $5 \times 5$ exploratory cross around that spot ($\pm 25$ units) to sense gradient direction. Phase-3 local search: from the best cross-centre run a small $3 \times 3$ pattern ($\pm 5$ units) and keep shrinking step/2 until improvement $< 1\%$. All coordinates rounded to 4 decimals; never re-sample the same point; budget 173 calls total.

**LLM$^\star$ Strategies in Rounds 2–21 (No Change)**

NA

*Phase summary: The strategy of the agent in rounds from 2 through 21 remained (NA).*

Figure 4: Kimi-K2 model in a sample run. The figure illustrates the agent's initial multi-phase plan: a coarse Latin Hypercube grid for global exploration, followed by local zoom-in and progressively finer refinements. However, subsequent rounds show no strategic adaptation (marked "NA"), indicating that the agent adhered rigidly to its original plan without incorporating feedback or diversification. This static approach highlights the limitations of non-adaptive strategies in complex search landscapes.

(2023) and ChatDev Qian et al. (2023) automate software development by assigning distinct roles to different agents. Multi-agent debate frameworks (such as MAD Liang et al. (2023), which employs a debate cycle among agents moderated by a judge agent, and Du et al. Du et al. (2024), where agents exchange answers to get a chance to modify their next responses) present another direction.

While these works follow Minsky's multi-agent view, our proposal, ACE takes a different path. ACE focuses on the reasoning process itself rather than focusing on how to delegate tasks to specialized agents or automate their communication. Great performance of ACE (as shown in section 4) suggests that the basic element of intelligence needs to include a Hegelian-inspired triad, not a single entity offering a complementary perspective to Minsky's multi-agent approach. Compared to debate-based proposals, from a philosophical qualitative perspective, as explained earlier in section A, there are key fundamental differences between debate and dialectics, which ACE draws inspiration from. Additionally, from a quantitative standpoint, our experiments and comparisons in section 4 highlight ACE's superior performance over debate-based works in SOP context.

## C  A COUPLE OF SAMPLES FOR LLM$^\star$ IN ACTION

Figures 4 and 5 demonstrate two samples of strategies used by LLM$^\star$ in separate runs, utilizing the Kimi-K2 and GPT-4.1 base models, respectively. In the first run (Figure 4), the agent lays out a clear multi-phase plan: begin with a broad, space-filling sweep over the full domain to avoid early clustering and surface promising regions; when a high value appears, trigger a local zoom using a small exploratory stencil to sense ascent directions; then switch to a compact pattern search that repeatedly shrinks its step size while guarding against re-sampling. After this initial plan, agent decides to not change its strategy in the subsequent rounds, (marked "NA"), and the run proceeds by repeatedly tightening the neighborhood around the incumbent best point until the query budget is exhausted. Overall, the strategy indicates a coarse-to-fine refinement strategy that is systematic and easy to interpret. However, the actual execution of the agent (through subsequent python codes) remains highly deterministic and lacks adaptive mechanisms. The execution rigidly follows a shrinking-grid pattern without introducing stochasticity, or any checks for diminishing returns. As a result, the agent commits early to local exploitation, never revisits other promising regions, and continues refining even when improvements become negligible. These issues stem from the absence of exploration–exploitation balancing logic in the strategy, leading to wasted queries and a high risk of getting trapped in a local maximum.

In the second example, (Figure 5), the GPT-4.1 model attempts to adapt its strategy based on the feedback it receives from the World. The agent follows a coarse-to-fine hierarchical grid strategy: it begins with broad exploration to map the landscape, then repeatedly zooms in around the single best point it has found, tightening the grid with each turn. The approach is deterministic and greedy, adapting to feedback only by narrowing the search window, without stochasticity, restarts,

**LLM⋆ Strategy in Turn 1**

We will begin with a coarse grid sampling approach to explore the entire search space. The range [-1000, 1000] is divided into a 5x5 grid, covering 25 evenly spaced points. This provides a preliminary understanding of the landscape.

**LLM⋆ Strategy in Turn 2**

Refine the search by exploring a finer grid around the current highest value point, (0.0, -1000.0), while ensuring all selected points remain within the valid range (-1000, 1000). This exploits the most promising region while maintaining local exploration.

**LLM⋆ Strategy in Turn 3**

Refine the search further by creating an even finer grid around the current highest value point (-100.0, -1000.0). This allows us to exploit the local neighborhood of the current best point and move toward the global maximum.

**LLM⋆ Strategy in Turn 4**

Focus on the region around (-125.0, -1000.0) with an even finer grid. This helps pinpoint the global maximum with greater precision.

**LLM⋆ Strategy in Turn 5**

Focus on the region around (-137.5, -1000.0) with an even finer grid to hone in on the peak with greater precision.

**LLM⋆ Strategy in Turn 6**

Narrow the search around (-143.75, -1000.0) with an even finer grid to pinpoint the exact location of the global maximum.

**LLM⋆ Strategy in Turn 7**

Focus the search on a very fine grid around (-146.875, -1000.0) to pinpoint the global maximum. Given the limited number of remaining queries, the grid will be extremely localized.

**LLM⋆ Strategy in Turn 8**

NA

Figure 5: A sample sequence of strategies employed by LLM⋆ over eight turns using the GPT-4.1 base model. The agent begins with a coarse global grid search and then repeatedly narrows its focus, applying finer grids around the current best point. While systematic, this deterministic zoom-in pattern lacks exploration–exploitation balance and fails to revisit other promising regions, ultimately leading to local entrapment and failure to locate the global optimum.

or diversification. While this yields a clean, interpretable trajectory and can work well on unimodal surfaces, it ultimately fails to find the global optimum here because it commits too early to local exploitation, spending most of its budget micro-refining a small neighborhood with diminishing returns. That said, locating and consistently refining a local maximum is noteworthy: it shows the agent recognized a regional peak and the qualitative relation between directional moves and observed changes in the objective. The core shortcomings are an inadequate exploration–exploitation balance, no revisiting of other promising regions from the initial scan, and no mechanisms to handle multiple peaks. A more adaptive design would reserve budget for continued exploration, introduce stochastic escapes and boundary-aware trust regions, or adopt global optimizers (e.g., BO, CMA-ES, etc.) that explicitly balance exploration and exploitation.

## D  A SAMPLE OF THE DIALECTICAL PROCESS IN ACE

Figures 6–8 illustrate how the dialectical method of ACE, (with the Kimi-K2 as base model), transforms the optimization process compared to earlier static strategies shown in Figures 4 and 5. Rather than committing to a single plan, the agent iterates through thesis, antithesis, and synthesis,

## Dialectical Progress in ACE (Part 1) — Turns 1–2

**Thesis 1**

Start with a two-stage hybrid search. Stage 1 (exploration): draw a low-discrepancy Latin Hypercube Sample (LHS) of 60 points that covers the whole 2000×2000 square. LHS spreads points evenly while leaving no large voids, giving a first rough map of peaks and valleys. Stage 2 (exploitation): after every LHS batch, fit a cheap surrogate (quadratic regression with interaction terms) to all observations, optimize it by brute-force on a 1%-resolution grid, and pick the 10 grid cells whose surrogate prediction is highest but whose centres have not been queried yet. Interleave these two stages until 173 queries are spent or the same best point is requested three times in a row (which will lock it as the believed global maximiser). All coordinates are rounded to four decimals to obey the rules.

**Antithesis 1**

**Critique:**

- 60 LHS points is a good first scan, but the surrogate model you plan (quadratic + interactions) is too weak for rugged landscapes; it will hallucinate smooth ridges and miss narrow spikes.
- The gap between best (1162.4) and second-best (907.3) is 255 units – large enough that a 5% tolerance window already reaches $\approx$1104. Do not chase the noise inside this window; instead verify whether the 1162 neighbourhood is a genuine peak or a fluke.
- No edge/corner check: the square corners are still untouched; many test functions hide extrema there.
- You are 60 / 175 queries in (115 left). At most 30 more should be spent on local polishing; the rest must go to uncovered regions.

**Issues to fix next:**

1. Immediately probe 8-points ring $\pm$25 around the current best to confirm curvature.
2. Add a space-filling set (e.g., 20 Sobol) biased toward the still-empty corners/edges.
3. Replace the quadratic surrogate by a flexible one (random-forest or RBF) before you start the "exploitation" stage.

**Synthesis 1 (Thesis 2)**

NA

**Antithesis 2**

**Critique:**

- You duplicated two neighbours (297.0339,-1000) and (347.0339,-1000); that burns 2 of your 115 remaining queries for zero information.
- The 3×3 stencil only moved 25 units; the jump from 1162 $\rightarrow$ 1180 shows the ridge continues east/south-east. Widen the next stencil to 50–75 units in that direction instead of re-sampling the same 25-unit cloud.
- Corner/edge probe was useful: (-1000,-1000)=627.9 is better than any previous edge value, but still 500 units below the current best – no need for dense edge packing yet.
- You still have not tested a surrogate; 80 points in, start a cheap RBF ensemble to tell you where prediction error is high, otherwise you will keep polishing locally while missing distant spikes.

**Issues ahead:**

1. 95% tolerance window is now $\geq$1121.7 – anything below that is irrelevant.
2. 95 queries left; allocate $\leq$15 more around the 1180 ridge, then launch a space-filling set (e.g., 30 Sobol) weighted toward unsurveyed quadrants.

Figure 6: Dialectical progress in ACE (Part 1) using the Kimi-K2 base model. The first thesis proposes a two-stage hybrid strategy: global exploration via Latin Hypercube Sampling (LHS) followed by surrogate-based exploitation. The antitheses critique weaknesses (simplistic surrogate, lack of edge coverage, premature local focus), setting the stage for more adaptive techniques in later turns.

using structured critique to refine its approach dynamically. The first thesis proposes a hybrid plan combining Latin Hypercube Sampling with a simple quadratic surrogate, but the antithesis immediately flags critical weaknesses—such as the surrogate's inability to capture rugged landscapes and the lack of edge coverage. This feedback drives subsequent syntheses toward more advanced

---

## Dialectical Progress in ACE (Part 2) — Turns 3–4

### Synthesis 2 (Thesis 3)

Keep two parallel tracks: A. Local: follow the rising ridge hinted by 1162→1177→1181. Use 50-unit steps E/SE; stop as soon as no improvement $\geq 0.5\%$ is seen for two consecutive rings. B. Global: spend the bulk of remaining queries on an adaptive RBF-guided search. Fit a thin-plate-spline RBF to all 80 observations, compute leave-one-out prediction error map on a coarse grid, and pick the 20 cells with highest upper-confidence-bound (UCB = pred + 2·std) that have not been queried. After every 20-point UCB batch, refit the RBF and repeat.

### Antithesis 3

**Critique:**

- Ridge march is working: 1181→1204 in 50-unit steps. Still no plateau, so keep pushing SE/E with 75-unit stride next.
- You wasted 25 pure-random Sobol points; most landed in already-saturated basins. Replace them with error-weighted space-filling.
- Duplicate filter is only local; you re-sampled (347.0339,-925.0) – costing 1 query. Use a global seen-set.
- 95% tolerance window is now $\geq 1143.6$; anything below can be ignored. Two values just crossed it: (560.66,-804.38,1170.8) and (324.28,859.22,985.4) – second ridge candidate.

**Issues next:**

1. 69 queries left; spend $\leq 12$ on the 1204 ridge, 15 on the new 1171 ridge, remainder on high-UCB cells.
2. Widen step to 75 units E/SE; if no +0.5% gain after two steps, declare local peak and switch to UCB-only.

### Synthesis 3 (Thesis 4)

Maintain global seen-set to avoid duplication. Local: continue ridge walk with 75-unit E/SE steps; stop when two consecutive rings give <0.5% gain. Global: build RBF on all 110 points, evaluate leave-one-out std on 100×100 grid, pick 20 cells with highest UCB. Interleave 12 local + 20 UCB per batch; 68 queries left.

### Antithesis 4

**Critique:**

- Ridge march still climbs: 1204→1218 at (472,-925). Step-size 75 is still productive; extend one more stride.
- You again sprinkled 20 uniform-random "UCB" points instead of using the actual RBF error surface.
- Duplicate guard worked, but you re-sampled the exact centre of the previous ring.
- 95% tolerance window is now $\geq 1157$. Only two new points cleared it; focus remaining queries on the 1218 ridge or true high-uncertainty zones.

**Issues / next moves:**

1. 48 queries left.
2. One last 75-unit E/SE extension; if gain < 0.5%, lock that ridge.
3. Fit RBF now, spend remaining 35 queries on top-UCB grid cells only.

Figure 7: Dialectical progress in ACE (Part 2). The agent transitions from static refinement to a two-track strategy that integrates Bayesian Optimization principles. Locally, it performs ridge climbing with adaptive step sizes to exploit strong gradients. Globally, it introduces an RBF surrogate model and uses the Upper Confidence Bound (UCB) acquisition function to guide exploration toward high-uncertainty regions. Antitheses highlight inefficiencies such as random sampling and delayed surrogate use, reinforcing the dialectical process as a mechanism for balancing exploration and exploitation.

techniques, including Radial Basis Function (RBF) surrogates, Upper Confidence Bound (UCB) acquisition, and explicit allocation of queries between local and global phases.

## Dialectical Progress in ACE (Part 3) — Turns 5–7

### Synthesis 4 (Thesis 5)

Build RBF surrogate on all 133 observations immediately; evaluate mean & std on 1% grid; pick 25 highest UCB cells not yet queried. Simultaneously probe the ridge one last time with 75-unit step east and north-east; if no $\geq 0.5\%$ gain, cease local march and devote all remaining queries to UCB-guided search.

### Antithesis 5

**Critique:**
- Ridge march stopped climbing: $1218 \rightarrow 1199$ (–1.5%). Peak is bracketed; stop local search.
- You again sampled 21 random grid cells instead of the 21 highest UCB cells; wasted 19 queries.
- You still have not fitted the RBF; without it you are blind to high-uncertainty zones.
- 27 queries left – too few for random sampling. Every remaining point must come from the surrogate's top-UCB list.

**Issues / next moves:**
1. Fit RBF now on all 155 observations.
2. Evaluate mean & std on the full grid, pick the top-27 UCB cells – no randomness.

### Synthesis 5 (Thesis 6)

NA

### Antithesis 6

**Critique:**
- RBF surrogate built, but you selected cells by prediction value, not UCB. All picks cluster near the 1218 peak.
- Every returned value sits inside the 95% tolerance window; confirms the peak but does not rule out distant spikes.
- 8 queries remain – too few to carpet-bomb the same neighborhood. You need one best-guess evaluation and then stop.

### Synthesis 6 (Thesis 7)

Given no additional queries remained, maximum is found at ...

Figure 8: Dialectical progress in ACE (Part 3). The agent attempts to consolidate its strategy by prioritizing UCB-guided sampling and enforcing duplicate filtering. While the synthesis nodes emphasize surrogate-driven global search, critiques reveal persistent execution flaws—such as selecting points by raw prediction instead of UCB and over-sampling near the current peak. These final turns underscore both the strengths and limitations of the dialectical approach: it fosters structured adaptation and diversification but still depends on correct implementation of its own prescriptions.

As the experiment progress, the strategy evolves into a two-track system: (i) local ridge climbing to exploit strong gradients and (ii) surrogate-guided global exploration using uncertainty estimates to target underexplored regions. This represents a significant improvement over the rigid, deterministic zoom-in patterns seen in previous runs. While some inefficiencies remain—such as delayed surrogate fitting and occasional random sampling—the dialectical process enforces diversification, duplicate filtering, and adaptive stopping rules, reducing wasted queries and mitigating local maximum traps. By embedding critique and synthesis into the reasoning loop, ACE achieves a more principled exploration–exploitation balance, leveraging Bayesian optimization principles without losing heuristic flexibility. This structured adaptability enables the agent to make better use of its query budget and substantially increases its chances of approaching the global optimum in complex, multimodal landscapes.

## E   MORE ON THE EVALUATIONS AND THE PROMPT TEMPLATES USED

**Prompt Templates:** Figure 9 shows the main template used for the LLM$^\star$ scheme. We use the same template for the main agent of other schemes compared in this paper, including ACE's Actor. Additionally, Figure 10 and 11 demonstrate the initial and transitional prompts used for ACE's Critic, respectively. The task of the Synthesizer, the Actor of the previous and next steps, will be identified through a transitional prompt, as shown in Figure 12.

**Majority Scheme:** To implement the Majority scheme and automate the solution, we use another agent called the poll worker. The poll worker checks different agents' responses and identifies the Majority response, which is the one with the highest consensus. Figure 13 shows the prompt template used for the poll worker. Unlike taking the majority vote after every agent completes the task in general scenarios, in our sequential decision-making problems, we need to take the majority vote in every round. Therefore, the poll worker processes the agents' responses at each round of interaction with the World, identifies the response with the majority consensus at each round so that the World can execute it and provide the feedback.

## Main Initial Prompt Template for LLM$^\star$ and Other Schemes

**Role Assignment**

*You are a great expert in the optimization topic and search algorithms.*

**Problem Definition**

You are tasked with examining an unknown function $f(x, y)$ ($-1000 \leq x, y \leq 1000$). You need to interact with the function $f(x, y)$ in order to locate the global maximum value. Here's how to do it:

1. *Define Your Strategy*: Start with creating a solid strategy to explore the space and solve the problem.

2. *Choose a Point (x, y)*: Based on your strategy, select unique NEW points $(x_1, y_1), (x_2, y_2), \ldots$ to evaluate the function.

3. *Get Feedback and Adjust Your Strategy*: After I reveal the values of $(x_1, y_1, f_1), (x_2, y_2, f_2), \ldots$ at your chosen points, adjust your strategy based on this feedback.

4. *Repeat the Process*: Continue this process for up to $QueryBudget$ queries (in the form of $(x_i, y_i)$) or until you are confident that you have found the global maximum.

*Note*: Finding a value in the range of $[0.95 \times (\text{Global Max}), \text{Global Max}]$ is equal to solving the problem.

**Response Format**

Here's how you should format your response:

- *MY_CURRENT_STRATEGY*: <explain your chosen strategy here>

- *MAX_SEEN_SO_FAR*: $x, y, f(x, y)$

- *NEXT*: <Python code snippet that generates the next coordinates and return a list of tuples [(xi, yi), ...]>

{...}

**General Rules and Examples of Acceptable/Unacceptable Responses**

Here are some rules that you must follow: {...}

**General Hints**

- The space is vast, and there will be several LOCAL maximums, so avoid choosing them as the answer. Make sure that you explore the space enough to ensure that your answer represents the GLOBAL maximum

- Asking for a certain coordinates multiple times, consumes your available remaining query budget and reduces your chances of finding the global maximum. So, utilize the responses so far to select only unique coordinates.

- A very important point is that YOU SHOULD NOT BE HASTY. You should be patient and explore the space thoroughly. However, remember that you have only a maximum of $QueryBudget$ queries to solve the problem.

{...}

**Examples of Function $f$**

Here are examples of function $f$ where it has multiple local maxima and one global maximum {...}

**Start Command**

Let's start. Create an excellent and efficient strategy and choose your first batch of coordinates accordingly.

Figure 9: Main initial prompt template used for LLM$^\star$ and other schemes

## Critic's Initial Base Prompt Template

**Role Assignment**

*You are a great expert in the optimization topic and search algorithms and will assist others in solving optimization problems.*

**Problem Definition**

Your task is to provide guidance, suggestions, and assistance to a very smart AI agent for solving an optimization problem.
The agent is to interact with an unknown function $f(x, y)$ $(-1000 \leq x, y \leq 1000)$ with the objective of identifying the global maximum value. Here is the procedure the agent will adhere to:

1. *Strategy Development*: The agent will begin by devising a comprehensive strategy to explore the space and tackle the problem.

2. *Point Selection (x, y)*: The agent will choose unique NEW points $[(x_1, y_1), (x_2, y_2), \ldots]$ for function evaluation, based on its strategy.

3. *Feedback Collection and Strategy Enhancement*: Once the values of $[(x_1, y_1, f_1), (x_2, y_2, f_2), \ldots]$ at the agent's selected points are revealed, the agent can refine its strategy using this feedback.

4. *Process Persistence*: The agent will continue this procedure for up to $QueryBudget$ queries (in the form of $(x_i, y_i)$) or until it is confident that the global maximum has been identified.

The agent should present its findings in the following way: {...}
The agent will comply with the following rules: {...}
Given this problem statement, your duty is to ensure that the agent identifies the global maximum value.

**General Guidelines**

To achieve your goal, please follow these guidelines:

1. After each step, you can critique the agent's chosen coordinates or its strategy. It is essential that you offer constructive criticism to improve its next moves.

2. It's important to remember that the space is vast, and there may be several LOCAL maximums, so you must help the agent avoid mistaking local maximum values for the answer {...}

3. You can offer suggestions and brainstorming to assist the agent in its task. Remember that the agent is very smart, so do not describe what the agent has already chosen or done! Limit your responses to constructive criticism.

4. Note: Finding a value in the range of $[0.95 \times (\text{Global Max}), \text{Global Max}]$ is equal to solving the problem, so discourage the agent to spend time on finding values that have small differences.

5. After every iteration, list the potential issues with the agent's strategy and decision so far.

6. Ensure that your responses are concise and to the point. Do not provide unnecessarily long responses.

{...}

**Agent's Response & the Corresponding Results**

Here is the Agent's response: $< Thesis_1 >$ and the corresponding results: $< Observations_1 >$

**Start Command**

Now, given all the info, review agent's response, make your criticism and suggestions, and detect potential issues ...

Figure 10: Initial prompt template used for the Critic in ACE

### Critic's Transitional Prompt Template

**Agent's Response & the Corresponding Results**

Here is the Agent's response: $< Thesis_i >$ and the corresponding results: $< Observations_i >$

**Start Command**

Now, given all the info, review agent's response, make your criticism and suggestions, and detect potential issues ...

Figure 11: Transitional prompt template used for the Critic in ACE

### Synthesizer's Prompt Template

**Synthesize Command**

The corresponding results are: $< Observations_i >$

To help you on your task, we provide you (the Agent/Actor) with the response from a reviewer who is observing your attempts:

$< Antithesis_i >$

Given the suggestions and comments provided, improve your strategy and continue.

Figure 12: The prompt template used for the Synthesizer in ACE. Note that Synthesizer is the Actor of the previous round, so it already has access to the $Thesis_i$. This provides an efficient handling of the context and token usage.

## Poll Worker Prompt Template

**Role Assignment**

*You are a great assistant with a strong background in AI and optimization problems.*

**Problem Definition**

You are assigned to work as a poll worker to analyze responses from multiple agents to a given problem. Each agent's response may include long sentences describing their strategy to solve the problem, code snippets, or other relevant information.

Your task is to identify the agent whose response is the most frequently specified among all agents. If there is a tie, you should randomly select one of the tied agents. Your response should only include the integer ID of the selected agent. Ensure that the selection process is fair and unbiased.

**General Guidelines**

- **Input Data**

  You will receive a list of responses from multiple agents. Each response is associated with a unique agent ID. *Example format:*

  - The response from agent $id_1$: $response_1$
  - The response from agent $id_2$: $response_2$
  - ...
  - The response from agent $id_n$: $response_n$

- **Processing**

  Analyze the responses to determine which agent's response is the most frequently specified. Evaluate the similarity of responses based on the nature of the answer, strategy, and major similarities, rather than exact wording. In case of a tie, randomly select one of the tied agents.

**Response Format**

Your output should be a single integer representing the ID of the agent with the most frequently specified response.
**Example output:** 3

**Examples**

Your response should only include the integer ID of the selected agent. You must avoid apologizing in your answers. Ensure that the selection process is fair and unbiased. *Example:* Given the following input:

- The response from agent 1: "Use a divide-and-conquer strategy to break the problem into smaller parts. Start with a few number of smaller parts"
- Agent #2: "Apply a divide-and-conquer approach to split the problem into manageable sections. Start with 10 parts"
- agent 3: "Implement a brute-force method to try all possible solutions"
- The Agent #4's response: "Use reinforcement learning to find the optimum solution"
- Agent 5: "Divide the problem into 10000 smaller parts and solve each part individually"

The most frequently specified strategy is "using divide-and-conquer with small number of total parts" which is provided by agents 1 and 2. Note that agent 5 specifies the divide-and-conquer part but with a large number of initial small parts. Therefore, you should output one of the IDs 1 or 2. If there is a tie, randomly select one of the tied IDs.
**Output:** 2

**Start Command**

Ok, let's start

Figure 13: The prompt template used for the poll worker agent in Majority scheme

