# OpenReview forum: "LLMs for Sequential Optimization Tasks: from Evaluation to Dialectical Improvement"
_ICLR.cc/2026/Conference — ICLR 2026 Conference Withdrawn Submission_

### Official Review · Reviewer_nxjG · 2025-10-28

**Soundness:** 4
**Presentation:** 3
**Contribution:** 4
**Rating:** 8
**Confidence:** 3

**Summary:**

The paper introduces **WorldGen**, a dynamic benchmark to test LLMs on **Sequential Optimization Problems (SOPs)** with controllable complexity. It finds that LLMs perform well on simple tasks but degrade as complexity increases. To improve this, the authors propose **ACE (Act, Critique, Evolve)**—a *dialectical reasoning framework* that enhances LLM problem-solving through iterative thesis–antithesis–synthesis cycles. Without retraining, ACE consistently outperforms baselines like Self-Reflection and Debate across multiple models and tasks.

**Strengths:**

*  ACE introduces a new reasoning paradigm grounded in dialectics, distinct from prompt-engineering or multi-agent scaffolds.
*  WorldGen allows scalable, contamination-free evaluation of LLMs in unseen optimization contexts.
* Seven LLMs across multiple reasoning baselines, with cost analyses and ablations.

**Weaknesses:**

* Current experiments use synthetic 3-D “worlds,” which may not represent real-world sequential optimization (e.g., reinforcement learning, control).
* The “Expert Solution” is human-crafted and not fully automated.
* While conceptually elegant, the Hegelian analogy may be seen as rhetorical rather than rigorously formalized.
* Some experimental details (e.g., number of iterations, random seeds) are omitted from the main text.

**Questions:**

* How does WorldGen scale to higher-dimensional or real-world tasks (e.g., scheduling, control)?
* Could ACE be formalized more concretely—for example, as an algorithmic update rule or meta-controller?
* Could the authors share the WorldGen generation code and parameters to ensure reproducibility?

---

### Official Review · Reviewer_gos8 · 2025-10-31

**Soundness:** 1
**Presentation:** 2
**Contribution:** 1
**Rating:** 2
**Confidence:** 4

**Summary:**

In this paper, the authors claim that large language models (LLMs) have potential for sequential optimization problems. They introduce a benchmark for generating such problems with controllable complexity to evaluate LLM performance. Furthermore, the authors propose a new framework (ACE) to enhance such performance.

**Strengths:**

This paper addresses a broad question and could serve as a good position paper if well executed (however, this is not the case; see weaknesses below). The related works section also serves as a good survey.

**Weaknesses:**

The paper suffers from unclear framing, vague methodology, and questionable meaningfulness. The notion of "sequential optimization problems", which is central to this paper's framing, is never clearly defined. Trying to decipher it leads to limited results. For example, the statement “the task naturally emerges as finding the maximum (or other extrema) in the generated n-dimensional world” provides no clear formulation of the objective, constraints, or difficulty of the problems being optimized. As a result, it is unclear what problem is actually being solved or how it relates to established optimization research.

Optimization is a broad field with well-defined benchmarks and taxonomies. Without a clear problem class or justification, the work lacks grounding. The proposed ACE is also insufficiently detailed: its mechanisms and implementation are not described in a way that enables reproduction or proper assessment. Such lack of clarity is not limited to a single method: overall, the reproducibility of this paper needs improvements.

As a result, it is infeasible to evaluate the real contribution of the manuscript. A large portion of the paper is devoted to introductory discussion and related work, while core technical content and empirical analysis are not well grounded. The paper reads more like a position paper rather than a concrete research contribution. While philosophy-inspired methods can be interesting and worth exploring, the work would need a clearer problem definition, rigorous benchmarking, and stronger experimental validation before it could be considered ready for publication.

**Questions:**

(See unclear aspects in the weaknesses section)

---

### Official Review · Reviewer_PmiA · 2025-11-01

**Soundness:** 2
**Presentation:** 4
**Contribution:** 1
**Rating:** 2
**Confidence:** 4

**Summary:**

The paper proposed a new task for llms, i.e. the optimization task. To systematically test across different LLMs, the paper proposes a new method for providing the optimization task without providing the background. Based on this, the paper proposed a WorldGen benchmark and test across 7 different models on the benchmark. It is found that the harder the task, the lower the performance LLMs are. To increase the performance, the paper proposed ACE, which effectively increase the LLMs' performance without retrainning or finetuning.

**Strengths:**

This paper is the first paper to my knowledge that applies LLMs on optimization tasks, and especially sequential optimization tasks.

**Weaknesses:**

1. The paper is self-contradictory that at the beginning the LLMs are treated as optimizers (get feedback from the environment and send out new queries). However, for the most part the paper, LLMs are designers, which uses different type of optimization methods.
2. Based on weakness 1, i think the paper is more on applying LLMs onto real world optimization tasks. I do not understand how given the function for X belongs to R^3 can be a good ecological validation of the method. And i highly disagree that such way of formalization the optimization problem does not exist in training regime (since this is what authors' want to achieve). The authors totally misunderstand the specific optimization question and the format of the optimization questions. Rather i would like to see more ecological experiments (like traveler salesman problem).
3. Although the paper mentions sequential optimization tasks, the optimization problem itself is not SOP (each action is based on previous actions). Instead, it is only the LLM states are self-dependent. Please modify the paper.
4. Hegelian-Dialectics has a special meaning of development byvercoming internal contradictions. I believe here it is a abuse usage of Hegelian-Dialectics. Please edit the paper.
5. The definition of budget also feels self-contradictory. It was defined as the queries towards the world. However, it is later defined as the number of tokens. I believe if number of tokens is the budget limit, then grid search is definitely the best way given infinite running time and running memory.

**Questions:**

What is the definition of complexity here?

---

### Official Review · Reviewer_fQM6 · 2025-11-01

**Soundness:** 2
**Presentation:** 3
**Contribution:** 2
**Rating:** 2
**Confidence:** 4

**Summary:**

The paper investigates whether off-the-shelf LLMs can serve as end-to-end designers for sequential optimization problems (SOPs). It introduces WorldGen, a procedurally generated benchmark of optimization 'worlds' with adjustable complexity, and ACE (Actor, Critic, Synthesizer), a dialectical inference-time loop that iteratively refines LLM-generated strategies without retraining. Experiments on synthetic 3-D worlds show ACE often increases success rates versus several prompting and multi-agent baselines, at the cost of higher token usage.

**Strengths:**

Addresses a timely, underexplored question: LLMs as autonomous designers under query budgets.

Proposes a practical, inference-time orchestration (Actor/Critic/Synthesizer) that requires no model fine-tuning.

Uses procedural instance generation (WorldGen) to mitigate contamination risks and scale difficulty.

Provides qualitative traces and prompt templates that clarify how ACE operates.

**Weaknesses:**

The details on WorldGen are insufficiently specified and not reproducible from the manuscript: missing generator code/pseudocode, families of functions, parameter distributions, formal criteria for L0/L1/L2, number of sampled worlds per level, and random seeds.

Key experimental details are missing: complete prompt templates and hyperparameters for baselines, LLM sampling settings (temperature/top-p), tokenization/token-counting, and the policy for executing model-generated Python (sandboxing, allowed libraries, timeouts, error handling).

No comparisons to classical black-box optimizers (e.g., Bayesian optimization with sensible kernels/acquisitions, CMA-ES, multi-start local search, random search) under identical query budgets; without these, practical utility is unclear.

Insufficient ablations: lacks experiments that isolate Actor / Critic / Synthesizer contributions, vary number of dialectical rounds, and control for total token/API budget (token-normalized baselines).

Statistical reporting is weak: small number of trials per world, few confidence intervals or significance tests; aggregate numbers may hide per-instance variability.

Limited scope: experiments are restricted to synthetic 3-D worlds; scalability to higher dimensions, noisy or constrained problems, and real-world SOPs is untested.

Security/safety concerns from executing arbitrary code are not addressed in detail (sandboxing, reproducibility, allowed libs), and broader misuse risks are not discussed sufficiently.

The philosophical (Hegelian) framing is evocative but adds little algorithmic novelty beyond existing critique-and-refine / multi-agent methods; superiority claims over debate-style schemes need stronger formal or empirical support.

**Questions:**

Please release WorldGen (code or detailed pseudocode) as part of the supplementary zip: exact families of functions, parameter distributions, formal definitions of L0/L1/L2, number of worlds per level, and random seeds used for reported experiments.

Provide the Expert Solution code and parameter settings used to set query budgets, with justification for why those budgets are fair baselines.

Publish exact prompt templates for every scheme and the LLM runtime settings (temperature, top-p, max-tokens), plus the method used to count/normalize tokens and calls.

Describe the Python execution environment: sandboxing approach, allowed libraries, timeouts, failure handling, and how execution errors were treated in scoring.

Add comparisons to classical optimizers (BO with sensible kernels and acquisition functions, CMA-ES, multi-start local search, random search) under identical query budgets and report success rates, queries-to-solution, and compute/token costs.

Provide ablations that (a) isolate each ACE component (Actor-only, Actor+Critic, Actor+Synthesizer), (b) sweep number of dialectical rounds, and (c) include token-normalized baselines (give single-agent extra tokens equal to ACE's total usage).

Increase statistical rigor: run more worlds and repeats, report standard errors/95% confidence intervals, and perform significance tests (e.g., paired bootstrap) for main comparisons.

Report wall-clock runtime and monetary cost (API-call) estimates in addition to token counts to assess practical feasibility.

Demonstrate or analyze scalability: results or analysis for higher-dimensional problems (n>3), noisy evaluations, and constrained domains, or clearly state limitations.

---

### Note · Authors · 2025-11-26

I have read and agree with the venue's withdrawal policy on behalf of myself and my co-authors.